# Antibody activities in hyperimmune plasma against the *Rhodococcus equi* virulence -associated protein A or poly-*N*-acetyl glucosamine are associated with protection of foals against rhodococcal pneumonia

Susanne K. Kahn[1], Colette Cywes-Bentley[2], Glenn P. Blodgett[3], Nathan M. Canaday[3], Carly E. Turner-Garcia[4], Mariana Vinacur[2], Sophia C. Cortez-Ramirez[1], Patrick J. Sutter[5], Sarah C. Meyer[5], Angela I. Bordin[1], Daniel R. Vlock[6], Gerald B. Pier[2], Noah D. Cohen[1]*

1 Department of Large Animal Clinical Sciences, Equine Infectious Disease Laboratory, College of Veterinary Medicine & Biomedical Sciences, Texas A&M University, College Station, TX, United States of America, 2 Department of Medicine, Brigham & Women's Hospital, Harvard Medical School, Boston, MA, United States of America, 3 *6666* Ranch, Guthrie, TX, United States of America, 4 Lazy E Ranch, Guthrie, OK, United States of America, 5 Mg Biologics, Inc., Ames, IA, United States of America, 6 Alopexx Inc., Concord, MA, United States of America

* ncohen@cvm.tamu.edu

## Abstract

The efficacy of transfusion with hyperimmune plasma (HIP) for preventing pneumonia caused by *Rhodococcus equi* remains ill-defined. Quarter Horse foals at 2 large breeding farms were randomly assigned to be transfused with 2 L of HIP from adult donors hyperimmunized either with *R. equi* (RE HIP) or a conjugate vaccine eliciting antibody to the surface polysaccharide β-1→6-poly-*N*-acetyl glucosamine (PNAG HIP) within 24 hours of birth. Antibody activities against PNAG and the rhodococcal virulence-associated protein A (VapA), and to deposition of complement component 1q (C′1q) onto PNAG were determined by ELISA, and then associated with either clinical pneumonia at Farm A (n = 119) or subclinical pneumonia at Farm B (n = 114). Data were analyzed using multivariable logistic regression. Among RE HIP-transfused foals, the odds of pneumonia were approximately 6-fold higher (P = 0.0005) among foals with VapA antibody activity ≤ the population median. Among PNAG HIP-transfused foals, the odds of pneumonia were approximately 3-fold (P = 0.0347) and 11-fold (P = 0.0034) higher for foals with antibody activities ≤ the population median for PNAG or C′1q deposition, respectively. Results indicated that levels of activity of antibodies against *R. equi* antigens are correlates of protection against both subclinical and clinical *R. equi* pneumonia in field settings. Among PNAG HIP-transfused foals, activity of antibodies with C′1q deposition (an indicator of functional antibodies) were a stronger predictor of protection than was PNAG antibody activity alone. Collectively, these findings suggest that the amount and activity of antibodies in HIP (*i.e.*, plasma volume and/or antibody activity) is positively associated with protection against *R. equi* pneumonia in foals.

**Data Availability Statement:** All relevant data are either within the manuscript and its Supporting Information files. The method for hyperimmunizing donors to produce plasma is proprietary and not available.

**Funding:** Funding for this work was provided by the Link Equine Research Endowment, Texas A&M University for the Infectious Disease Epidemiology program. The Link Endowment provides programmatic support on an annual basis to me, and there is not a grant number associated with this internal funding. I am also supported by an endowed chair position, the Patsy Link Chair in Equine Research. Drs. Glenn Blodgett and Nathan Canaday are employees of the 6666 Ranch and Dr. Carly Turner-Garcia is employed by the Lazy E Ranch. Sarah Meyer and Patrick Sutter are employed by Mg Biologics, Inc. and Dr. Daniel Vlock is employed by ALOPEXX Vaccines. The 6666 Ranch, Lazy E Ranch, Mg Biologics, and ALOPEXX provided support in the form of salaries, but did not have any additional role in the study design, data collection and analysis, decision to publish, or preparation of the manuscript.

**Competing interests:** I have read the journal's policy and the authors of this manuscript have the following competing interests: Gerald B. Pier (GBP) is an inventor of intellectual properties [human monoclonal antibody to PNAG and PNAG vaccines] that are licensed by Brigham and Women's Hospital to Alopexx Vaccine, LLC, an entity in which GBP also holds equity. As an inventor of intellectual properties, GBP also has the right to receive a share of licensing-related income (royalties, fees) through Brigham and Women's Hospital from Alopexx Vaccine, LLC. GBP's interests were reviewed and are managed by the Brigham and Women's Hospital and Partners Health care in accordance with their conflict of interest policies. Colette Cywes-Bentley (CC-B) is an inventor of intellectual properties [use of human monoclonal antibody to PNAG and use of PNAG vaccines] that are licensed by Brigham and Women's Hospital to Alopexx Vaccine, LLC. As an inventor of intellectual properties, CC-B also has the right to receive a share of licensing-related income (royalties, fees) through Brigham and Women's Hospital from Alopexx Vaccine, LLC. Daniel R. Vlock is the Chief Executive Officer of ALOPEXX Vaccines and owns the rights to the PNAG vaccine. Dr. Vlock had no role in the design or analysis of the study. SC Meyer and PJ Sutter work for MG Biologics that produced the plasma and thus might have potential earnings; however, they had no part in the design or analysis of the study. Drs. Glenn Blodgett and Nathan Canaday are

# Introduction

*Rhodococcus equi* (*R. equi*) is a common cause of severe pneumonia in foals [1–5]. Virulent strains of this facultative, intracellular pathogen contain a plasmid that encodes for the virulence-associated protein A (VapA) that is necessary for bacterial replication in macrophages [6]. Pneumonia caused by *R. equi* is endemic at many horse-breeding farms, with annual cumulative incidence at farms often affecting 20% to 40% of the foal population [7–9]. At endemic farms, costs can be high for treatment, veterinary care, long-term therapy, and lost revenue from deaths of foals infected with *R. equi*. In addition to these immediate costs, *R. equi* pneumonia has a long-term detrimental effect to the equine industry because foals that recover from the disease are less likely to race as adults [10].

Pneumonia caused by *R. equi* is recognized as either clinical or subclinical forms [11,12]. The clinical form of *R. equi* pneumonia has an insidious progression: pathological changes in the lungs are well-advanced by the time clinical signs develop [3,4,6]. The subclinical form of *R. equi* pneumonia is characterized by the presence of pulmonary consolidations or abscesses identified by thoracic ultrasonography performed as a screening test at endemic farms in the absence of overt clinical signs of pneumonia [11,12]. Foals with ultrasonographically-identified pulmonary lesions greater than a certain threshold of a maximal diameter (*e.g.*, $\geq 2$ cm of maximum diameter) but lacking other clinical signs are often treated with antimicrobials [12]. The rationale for this screen-and-treat approach is that it will reduce mortality and duration of treatment of foals at endemic farms [12].

Methods for preventing *R. equi* pneumonia include chemoprophylaxis, vaccination, and administration of hyperimmune plasma (HIP) [13–30]. Of these, the only USDA-approved and well-established method for reducing the incidence of *R. equi* pneumonia is transfusion of HIP from equine donors hyperimmunized against *R. equi* (RE HIP) [13–15,17]. In addition to RE HIP, we recently demonstrated that transfusing foals between 12 to 24 hours after birth with plasma from donors hyperimmunized against the bacterial capsular polysaccharide β-1→6-poly-*N*-acetyl glucosamine (PNAG) prevented *R. equi* pneumonia following intra-bronchial infection at age ~28 days, whereas transfusion with commercial plasma from donor horses that were not hyperimmunized against either PNAG or *R. equi* and that had only background levels of antibody activity against PNAG and VapA failed to protect foals similarly infected [31]. Additionally, our laboratory has shown that PNAG HIP is superior to both RE HIP and standard plasma at mediating opsonophagocytic killing of *R. equi* by equine neutrophils [32]. Fixation of the complement component 1q (C'1q) to the PNAG antigen with vaccination-derived antibodies is considered essential to the functional activity of these antibodies both *in vitro* and within sera of foals receiving anti-PNAG antibodies via passive transfer from vaccinated dams [31,32].

Evidence of the effectiveness of HIP for reducing the incidence of *R. equi* pneumonia under field conditions, however, remains variable and conflicting, [12–16,28,33] and in the case of PNAG HIP is lacking. One possible explanation for the irregular effectiveness of RE HIP under field conditions is variable dosing. Results of observational studies indicate that administration of 2 L of RE HIP to foals is superior to administration of 1 L for reducing the cumulative incidence of clinical or subclinical pneumonia [29,30]. Moreover, the activity of *R. equi*-specific antibody varies among manufacturers and among lots/batches within manufacturers [34]. Collectively, these findings indicate that variation in the amount of antibody transfused to a foal is inversely related to the risk of pneumonia developing in that foal. Specific evidence of an association between antibody activities in transfused foals and protection against pneumonia, however, is limited. Thus, we conducted a randomized, controlled, double-masked field trial to examine the association between disease outcome and relative antibody activities

employees of the 6666 Ranch and Dr. Carly Turner-Garcia is employed by the Lazy E Ranch. Sarah Meyer and Patrick Sutter are employed by Mg Biologics, Inc. and Dr. Daniel Vlock is employed by ALOPEXX Vaccines. The 6666 Ranch, Lazy E Ranch, Mg Biologics, and ALOPEXX provided support in the form of salaries, but did not have any additional role in the study design, data collection and analysis, decision to publish, or preparation of the manuscript. This does not alter our adherence to PLOS ONE policies on sharing data and materials.

(*i.e.*, ratio of optical density [OD] of sample to OD of positive control) to the virulence associated protein A (VapA) of *R. equi* and PNAG, and activity of deposition of C'1q onto PNAG among foals randomly assigned to be transfused with 2 L of either RE HIP or PNAG HIP at 2 large breeding farms where *R. equi* pneumonia is endemic and where transfusion of RE HIP was historically used to control *R. equi* pneumonia. These farms differed in their diagnostic approach: Farm A did not use screening to identify foals prior to the onset of clinical signs (*i. e.*, diagnosis of *R. equi* pneumonia was based on detecting clinical signs of pneumonia), whereas Farm B used a combination of results of thoracic ultrasonographic screening and complete blood counts (CBC) to identify foals with pulmonary lesions and abnormal findings of CBC for presumptive diagnosis of subclinical *R. equi* pneumonia. We hypothesized that the cumulative incidence of clinical and subclinical *R. equi* pneumonia would be significantly lower either among foals transfused with RE HIP that had higher relative antibody activities to VapA, or among foals transfused with PNAG HIP that had higher antibody activities to PNAG or C'1q deposition onto PNAG.

## Materials and methods

### Study population

The study was approved by Texas A&M University's Institutional Animal Care and Use Committee and the Clinical Research Review Committee of the Texas A&M University's College of Veterinary Medicine & Biomedical Sciences (Animal Use Protocol 2018–0429), and included signed informed consent from either the owner or agent of the owner for all study foals. The study was conducted during the 2019 foaling season and included foals from 2 large breeding farms (Farm A and Farm B) that had a history of cumulative incidence of *R. equi* pneumonia ≥ 20% per foaling season over the preceding 5 years, and at which RE HIP was used historically to control pneumonia caused by *R. equi*. Each farm was known to have >150 foals born annually that resided through weaning at the farm. Diagnosis of presumed *R. equi* pneumonia among foals at Farm A was made on the basis of clinical signs (*i.e.*, **clinical pneumonia**), whereas diagnosis of presumed *R. equi* pneumonia at Farm B was made on the basis of results of thoracic ultrasonographic screening and specific abnormal findings of CBCs (*i.e.*, **subclinical pneumonia**). These farms were intentionally selected to allow us to examine the association of serum activity against antigens of interest with pneumonia among foals diagnosed with either clinical or subclinical pneumonia because both approaches are commonly used in private equine practice [35]. To be eligible for inclusion in the study, foals were required to have been healthy at birth and to have evidence of adequate passive transfer of immunoglobulins based on a commercial test kit (SNAP Foal IgG test, IDEXX, Inc.). At each farm, a total of 120 healthy Quarter Horse foals (n = 240 total foals) were randomly assigned in equal numbers using a blocked design to 1 of 2 groups: Group 1 received PNAG HIP (n = 60 per farm) and Group 2 received RE HIP (n = 60 per farm). This sample size was based on the number of foals available at each farm, and the number of liters of PNAG HIP available to the investigators. To the authors' knowledge, published data regarding the distribution of specific antibody levels immediately post-transfusion among foals transfused with either RE HIP or PNAG HIP were not available for *a priori* sample size calculations.

### Transfusions and clinical evaluation

Each foal was transfused with 2 L (approximately 40 mL/kg of body weight) of plasma within 24 hours of birth by trained veterinary technical staff or veterinarians at each farm. Foals were transfused with either RE HIP or PNAG HIP from a single manufacturer (Mg Biologics, Inc., Ames, IA). The RE HIP was derived from donor horses hyperimmunized using a propriety

method against *R. equi* and the PNAG HIP was derived from donor horses hyperimmunized, using a propriety method, with a conjugate vaccine composed of pentamers of β-1-6-linked glucosamine covalently linked to tetanus toxoid as a carrier protein (5GlcNH$_2$-TT) [31]. Plasma was labeled by the manufacturer as either 1 or 0 in order to mask the identity of the plasma both to those individuals transfusing foals at farms and to those performing data analysis. Treatment order (*i.e.*, allocation sequence) at each farm was pre-assigned randomly by investigators at Texas A&M University prior to the foaling season based on expected foaling dates of mares, and this treatment order was sent to the farm veterinarians prior to initiation of the study. Serum samples (4 mL) were collected immediately post-transfusion from the jugular vein contralateral to the jugular vein used for transfusion. These sera were used to determine relative antibody activities in the foals' sera as described below.

At Farm A, foals were monitored by the farm veterinary medical and veterinary technical staff at least twice daily for signs of clinical pneumonia. These signs included lethargy, coughing, depressed attitude, increased respiratory rate (> 60 breaths/minute) or effort (abdominal lift, flaring nostrils), and extra-pulmonary manifestations of *R. equi* infection such as polysynovitis or uveitis. Foals were diagnosed with presumed *R. equi* pneumonia if they had all of the following: 1) cough; 2) fever (rectal temperature > 39.4˚C); 3) lethargy or tachypnea or dyspnea; and 4) ultrasonographic evidence of pulmonary abscess(es) or consolidation(s) ≥ 2 cm in maximal diameter. Any foals found to have clinical signs of pneumonia were tested by complete blood count (CBC) and thoracic ultrasonography performed by the farm veterinarians. As noted previously, the veterinarians diagnosing the foals were masked to the identity of the plasma transfused to individual foals. Foals that developed pneumonia were treated with clarithromycin (7.5 mg/kg; orally; q 12 h) and rifampicin (5 mg/kg; orally; q 12 h) until resolution of clinical signs and thoracic ultrasonographic lesions. Medical records, including reports of all findings and treatments, were maintained daily for each individual foal.

At Farm B, thoracic ultrasonography was performed on all foals at ages 5, 7, and 9 weeks by farm veterinarians to examine the lungs for pulmonary abscesses or consolidations. If foals had consolidations or abscesses ≥ 2 cm in maximal diameter and increased concentrations of white blood cells, neutrophils, or fibrinogen detected from results of CBCs performed concurrently with thoracic ultrasound screening examinations, they were treated for presumed subclinical *R. equi* pneumonia with azithromycin (10 mg/kg; orally; q 24 h) and rifampicin (5 mg/kg; orally; q 12 h) until resolution of thoracic ultrasonographic lesions. Medical records, including reports of all findings and treatments, were maintained daily for each individual foal.

At the end of the season, the medical records from both farms were reviewed, and the proportion of foals that developed pneumonia attributed to *R. equi* (either clinical at Farm A or subclinical at Farm B) was determined. Diagnosis of *R. equi* pneumonia was determined prior to data analysis, and data analysis was performed prior to the unmasking of plasma type. In addition to the health data, the following information was collected for each foal: 1) date of birth; 2) sex; 3) results of semiquantitative blood concentration of immunoglobulin G; 4) whether there was a reaction to transfusion with hyperimmune plasma.

## Immunoglobulin ELISA

Serum samples from study foals were tested by enzyme-linked immunoassay (ELISA) for relative activities of antibodies against PNAG and the VapA protein of *R. equi*. ELISA plates (Maxisorp, Thermo Scientific, Rochester, NY, USA) were coated with either 0.6 μg/ml of purified PNAG or 0.5 μg/ml purified VapA diluted in sensitization buffer (0.04M PO4, pH 7.2) overnight at 4˚C [31]. Plates were washed 3 times with PBS containing 0.05% Tween 20, blocked

with 120 μl of PBS containing 1% skim milk for 1 hour at 37°C, and washed again. Foal serum samples (100 μl) were added in duplicate to wells of the ELISA plate and incubated for 1 hour at 37°C. Serum samples were initially diluted in the incubation buffer (PBS with 1% skim milk and 0.05% Tween 20) to 1:100. A sample each of PNAG HIP and of RE HIP were included in each ELISA plate as controls: for the PNAG ELISA, the PNAG HIP was the positive control and the RE HIP was the negative control, and for the VapA ELISA the RE HIP was the positive control and the PNAG HIP was the negative control. Plates were washed again, then 100 μl per well of anti-horse IgG conjugated to HRP (Bethyl Laboratories, Montgomery, TX, USA, diluted at 1:30,000) was added to the wells. Plates were incubated for 1 hour at room temperature and then washed again. SureBlue Reserve One Component TMB Microwell Peroxidase Substrate (SeraCare, Gaithersburg, MD, USA) was added to the wells for 2 minutes. The reaction was stopped by adding sulfuric acid solution to the wells. Optical densities (ODs) were determined at a wavelength of 450 nm by using a microplate reader. The relative activity of antibody was calculated by dividing the individual sample OD values by that of the respective positive control from the same plate, which we defined as the **OD ratio**.

### C′1q deposition assays

The rationale for testing deposition of C′1q onto PNAG is that it is a functional assay: anti-PNAG antibodies require complement deposition to mediate their opsonic killing, and not all antibodies to PNAG fix complement [31,32]. An ELISA targeting the C′1q component of C′1 (C′1q) was used to determine the serum endpoint activities for deposition of equine C′1 onto purified PNAG. ELISA plates were sensitized with 0.6 μg PNAG/ml and blocked with skim milk as described above. Dilutions of different foal sera were added in 50-μl volumes, after which 50 μl of 10% intact, normal horse serum were added as a source of C′1q. After 60 minutes of incubation at 37°C, plates were washed and 100 μl of goat anti-human C′1q, which also binds to equine C′1q, diluted 1:1,000 in incubation buffer was added and plates were incubated at room temperature for 60 minutes. After washing, 100 μl of rabbit anti-goat IgG whole molecule conjugated to alkaline phosphatase and diluted 1:1,000 in incubation buffer was added, and then incubated for 1 hour at room temperature. Washing and developing of the color indicator was then performed by adding p-nitrophenyl phosphate substrate and color development determined after 60 min at room temperature. $OD_{405}$nm values of this highest serum dilution tested were used to determine relative activity. Negative OD values after background subtraction (no primary antibody added) indicated sera with less activity than this control [31].

### Data analysis

We first compared the OD ratios of VapA, and PNAG and OD activities for C′1q deposition onto PNAG between foals transfused with RE HIP and those transfused with PNAG HIP using the generalized linear modeling (glm) function in R (version 3.6.1) and an identity link, with OD ratios (for VapA and PNAG) or relative OD (for C′1q) as the dependent (outcome) variable and plasma group as the independent variable. The purpose of this analysis was to ensure that antibody activities differed significantly between groups as expected (*e.g.*, significantly higher VapA antibody OD ratios among foals transfused with RE HIP than among foals transfused with PNAG HIP), as a measure of internal validity of the study. The primary questions of interest for this study were whether the OD ratios of VapA *among RE HIP-transfused foals* were significantly lower among foals that developed pneumonia, and whether the OD ratios against PNAG and or relative OD of C′1q *among PNAG HIP-transfused foals* were significantly lower among foals that developed pneumonia. Data were analyzed using

multivariable logistic regression within the glm function in R using a logit link, with pneumonia as the binary outcome variable and relative antibody activity level for a given antigen and farm as dependent variables. For purposes of analysis, antibody activities were analyzed as a binary variable using the median OD ratio/relative OD among foals transfused with a given plasma (*e.g.*, median VapA OD ratio among foals transfused with RE HIP). The median was used for purposes of simplicity of analysis, and was not selected as a diagnostic cut-point for protection against pneumonia. Farm was included in the models to account for the potential effects of differences between farms in the method of diagnosis. Although not a primary aim of the study, we also compared the 2 different plasmas (RE HIP and PNAG HIP) for protection against pneumonia using multivariable logistic regression within the glm function in R with logit link, with pneumonia as the binary outcome and plasma type and farm as dependent variables. All data analysis was performed using the R program (Version 3.6.1, R Core Team, Vienna, Austria). Significance for all analyses was set at $P < 0.05$, and 95% confidence intervals (95% CI) were estimated using maximum likelihood methods.

## Results

### Study population

Farm A had a total of 119 foals included in the project; 60 foals were transfused with RE HIP and 59 were transfused with PNAG HIP. One foal at Farm A was lost to follow up due to complications associated with neonatal isoerythrolysis. Data from this foal was excluded from analysis. Farm B had a total of 114 foals included; 57 foals were transfused with RE HIP (plasma 0), and 57 were transfused with PNAG HIP (plasma 1). Six of the 120 eligible foals from Farm B were excluded because of an unanticipated shortfall of plasma production by the manufacturer. Among foals at both farms transfused with RE HIP, the median OD ratio of VapA antibodies was 0.88 (range, 0.64 to 1.09). Among foals at both farms transfused with PNAG HIP, the median OD ratio of PNAG antibodies was 0.54 (range, 0.29 to 0.75) and the median relative OD $_{405}$nm for C′1q deposition was 1.01 (range, 0.07 to 2.99). These median values were used as cut-points to create binary variables (*i.e.*, high versus low) for antibody levels for use in logistic regression modeling. At both farms, OD ratios of VapA antibodies were significantly ($P < 0.05$) higher among foals transfused with RE HIP than with PNAG HIP (Fig 1). Similarly, the OD ratios for PNAG antibodies and relative OD values for C′1q deposition were significantly ($P < 0.05$) higher among foals transfused with PNAG HIP than RE HIP (Figs 2 and 3). None of the foals at either farm was noted to have an adverse reaction to transfusion. No deaths of foals at either farm were attributed to *R. equi* pneumonia.

### VapA antibody OD ratios following RE HIP transfusion

Of the 233 foals from both farms, 117 foals were transfused with RE HIP. Of those 117 foals transfused with RE HIP, 29% (34/117) developed either clinical or subclinical pneumonia and 71% (84/117) remained healthy. For logistic regression modeling, a low level of antibody activity against VapA was defined as an OD ratio $\leq 0.89$ (the population median) and a high level of antibody activity against VapA was defined as OD ratio $> 0.89$. The proportion of foals that developed pneumonia among foals with a low level of VapA activity was 40% (24/60), whereas the proportion of foals that developed pneumonia among foals with a high level of VapA activity was 18% (10/57). The proportion of foals transfused with RE HIP that developed pneumonia at Farm A was 42% (25/60) whereas the proportion that developed pneumonia at Farm B was only 16% (9/57). Using multivariable logistic regression analysis of the RE HIP-transfused foals, the odds of pneumonia were approximately 6-fold higher ($P = 0.0005$) among foals with a low level of VapA antibody activity relative to foals with a high level of VapA antibody

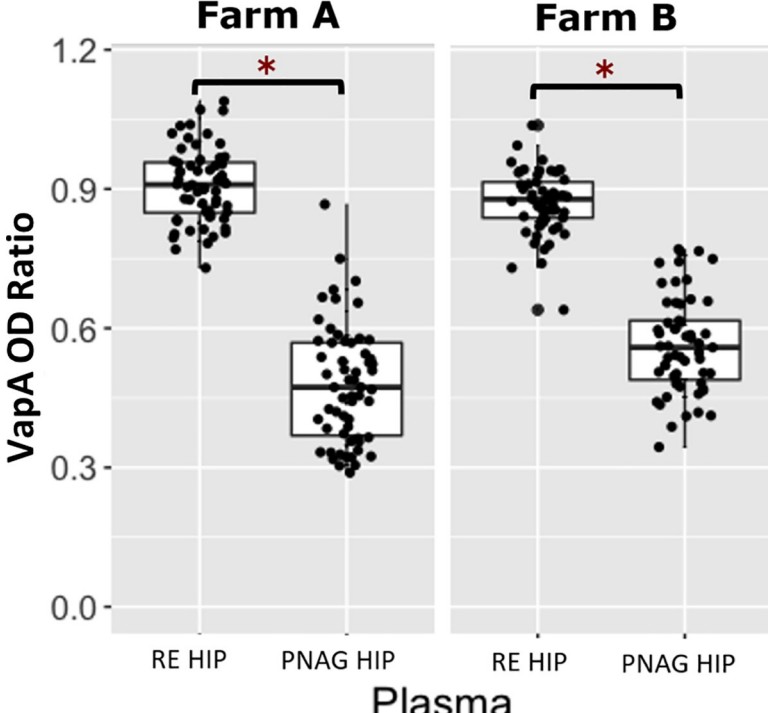

**Fig 1. Comparison of VapA optical density ratios between *R. equi* hyperimmune plasma and PNAG hyperimmune plasma.** Boxplots of optical density (OD) ratios from ELISAs measuring deposition onto purified VapA protein by serum antibodies in 119 foals from Farm A and 114 foals from Farm B stratified by plasma type, faceted by farm. Foals transfused with *R. equi* hyperimmune plasma (RE HIP) at both farms had significantly higher (asterisks represent P<0.05) VapA OD ratios than foals that were transfused with PNAG hyperimmune plasma (PNAG HIP). The mean (95% CI) OD ratio of anti-VapA antibodies among foals at Farm A and B were 0.91 (0.89 to 0.93) and 0.87 (0.85 to 0.89), respectively among foals transfused with RE HIP and were 0.48 (0.45 to 0.52) and 0.56 (0.54 to 0.59), respectively, among foals transfused with PNAG HIP.

activity, accounting for effects of farm (Fig 4, Table 1). Using multivariable logistic regression, the odds of pneumonia among foals transfused with RE HIP were approximately 7-fold higher (P = 0.0002) for foals at Farm A than for Farm B, accounting for effects of VapA antibody activity (Fig 4, Table 1).

## PNAG antibody OD ratios following PNAG HIP transfusion

Of the 233 foals from both farms, 116 were transfused with PNAG HIP. Of the 116 foals transfused with PNAG HIP, 22% (25/116) developed pneumonia. For logistic regression modeling, a low level of antibody activity against PNAG was defined as an OD ratio ≤ 0.54 (the population median) and a high level of antibody activity against PNAG was defined as an OD ratio > 0.54. The proportion of foals that developed pneumonia among foals with low PNAG antibody activity was 31% (18/58), whereas the proportion of foals with pneumonia among foals with high level of antibody activity against PNAG OD ratios was 12% (7/58). Among foals transfused with PNAG HIP, at Farm A 27% (16/59) developed pneumonia compared with 16% (9/57) of foals at Farm B. Using multivariable logistic regression analysis of the PNAG HIP-transfused foals, the odds of pneumonia were approximately 3-fold higher (P = 0.0005) among foals with a low level of antibody activity against PNAG relative to foals with a high level of antibody activity against PNAG, accounting for effects of farm; the odds of pneumonia, however, did not differ significantly (P = 0.4174) between farms (Fig 5, Table 2).

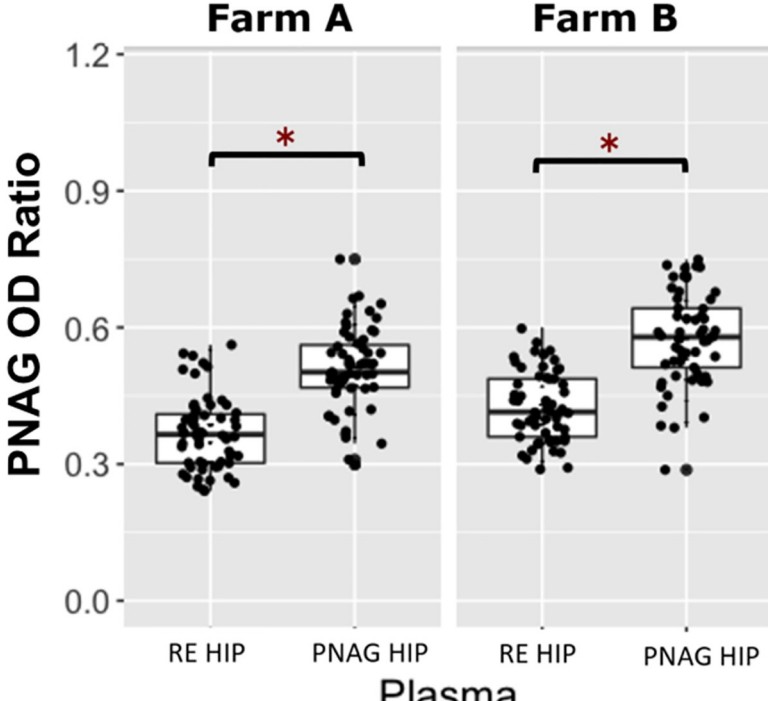

**Fig 2. Comparison of PNAG optical density ratios between *R. equi* hyperimmune plasma and PNAG hyperimmune plasma.** Boxplots of optical density (OD) ratios from ELISAs measuring deposition onto PNAG by serum antibodies in 119 foals from Farm A and 114 foals from Farm B, stratified by plasma type, faceted by farm. Foals transfused with PNAG hyperimmune plasma (PNAG HIP) at both farms had significantly higher (asterisks represent P < 0.05) OD ratios for PNAG than foals that were transfused with *R. equi* hyperimmune plasma (RE HIP). Statistical significance is indicated by asterisks. The mean (95% CI) OD ratio of anti-PNAG antibodies among foals at Farm A and B were 0.51 (0.49 to 0.53) and 0.58 (0.55 to 0.60), respectively among foals transfused with PNAG HIP and were 0.37 (0.35 to 0.39) and 0.43 (0.40 to 0.43), respectively, among foals transfused with PNAG HIP.

## C′1q activity

For logistic regression modeling, a low level of C′1q deposition activity was defined as a relative $OD_{405}$nm activity of $\leq 1.01$ (the population median) and a high activity level was defined as a relative $OD_{405}$nm activity $> 1.01$. The proportion of foals that developed pneumonia among those with low C′1q activities was 39% (22/57), whereas the proportion of foals with pneumonia among foals with high C′1q activities was 5% (3/57). As noted above, at Farm A, 27% (16/59) of foals transfused with PNAG HIP developed pneumonia compared with 16% (9/57) of foals receiving PNAG HIP at Farm B. Using multivariable logistic regression analysis of the PNAG HIP-transfused foals, the odds of pneumonia were approximately 11-fold higher (P = 0.0003) among foals with low C′1q activities (*i.e.*, $\leq 1.01$) relative to foals with high activities for C′1q deposition; the odds of pneumonia, however, did not differ significantly (P = 0.9777) between farms (Fig 6, Table 3).

## Association of plasma type with pneumonia

Of the 233 foals from both farms, 117 were transfused with RE HIP and 116 were transfused with PNAG HIP. The proportion of foals that developed pneumonia was 29% (34/117) among foals transfused with RE HIP and 21% (25/116) among foals transfused with PNAG HIP (Fig 7). Using multivariable logistic regression with pneumonia as the binary outcome and plasma type and farm as dependent variables, the odds of pneumonia in foals transfused with RE HIP

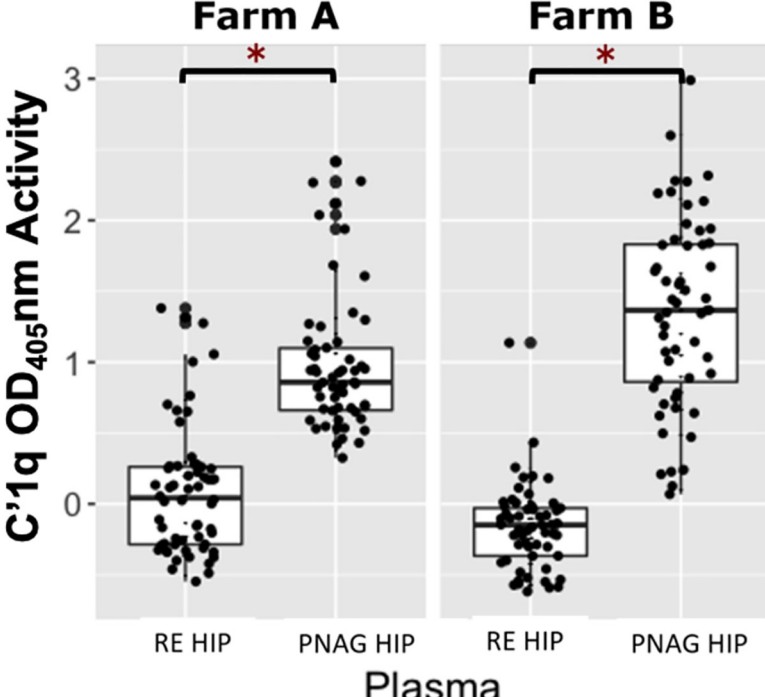

**Fig 3. Comparison of C′1q relative optical density between *R. equi* hyperimmune plasma and PNAG hyperimmune plasma.** Boxplot of relative optical density (OD$_{405}$nm) activities for deposition of complement component 1 (C′1q) onto PNAG by serum antibodies in 119 foals from Farm A and 114 foals from Farm B, stratified by plasma type, faceted by farm. Foals transfused with PNAG hyperimmune plasma (PNAG HIP) at both farms had significantly higher (asterisks represent P < 0.05) C′1q OD$_{405}$nm activities than foals that were transfused with *R. equi* hyperimmune plasma (RE HIP). The mean (95% CI) OD activity of C′1q antibodies among foals at Farm A and B were 0.98 (0.85 to 1.11) and 1.35 (1.17 to 1.53), respectively among foals transfused with PNAG HIP and were 0.09 (-0.03 to 0.21) and -0.17 (-0.24 to 0.09), respectively, among foals transfused with PNAG HIP.

were not significantly higher among foals transfused with PNAG HIP (OR = 1.5, 95% CI, 0.82 to 2.79; P = 0.1832) relative to foals transfused with RE HIP, adjusted for effects of farm. However, the odds of pneumonia were approximately 2.8-fold higher (P = 0.0013) for foals at Farm A than for foals at Farm B (Table 4): 34% (41/119) of foals at Farm A developed pneumonia compared to 16% (18/114) at Farm B (Fig 8).

## Discussion

The odds of developing either clinical or subclinical *R. equi* pneumonia were inversely associated with antibody activities against VapA among foals transfused with RE HIP, and with antibody activities against PNAG and C′1q among foals transfused with PNAG HIP. The validity of these results is supported by prior studies demonstrating that antibodies to PNAG protect against experimental intrabronchial infection of foals with *R. equi*, [31] and evidence that plasma with high relative antibody activity against VapA is protective against *R. equi* infection [13,36,37]. These findings are important for equine veterinarians and equine farm managers because they provide further evidence of the effectiveness of transfusion of RE HIP and PNAG HIP to protect foals against *R. equi* pneumonia and because HIP remains the only USDA-approved method for controlling *R. equi* pneumonia at endemic equine breeding farms.

The finding that VapA and PNAG antibody levels and the relative deposition of C′1q onto PNAG appears to be a useful correlate of protective immunity and could be an important

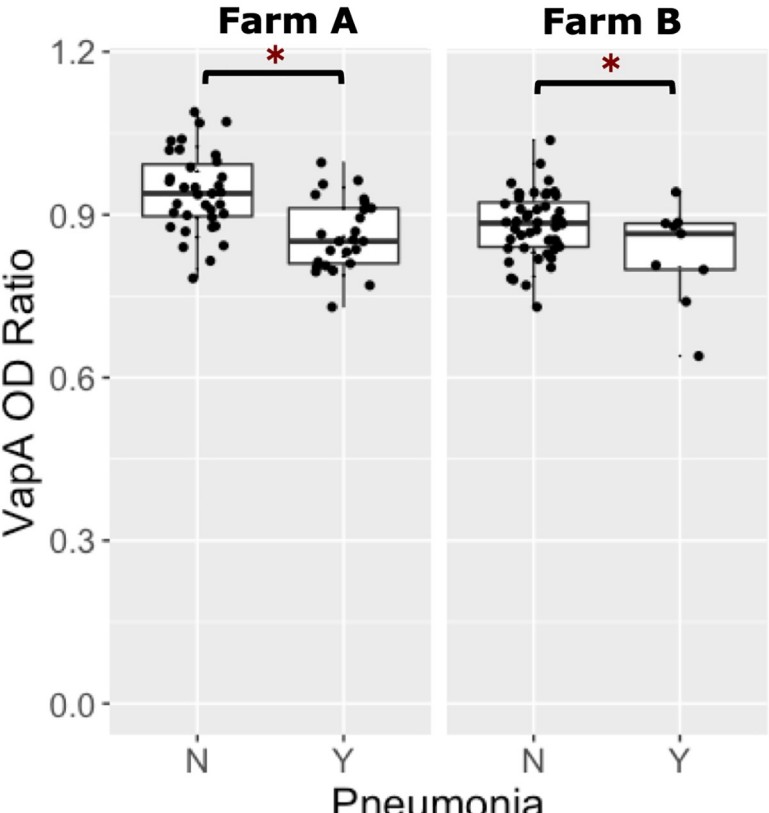

**Fig 4. Foals transfused with *R. equi* hyperimmune plasma and their comparison of VapA optical density ratios with whether they developed pneumonia or remained healthy.** Boxplots of optical density (OD) ratios from ELISAs measuring deposition onto purified VapA protein by serum antibodies in 117 foals transfused with *R. equi* hyperimmune plasma (RE HIP) stratified by pneumonia status, faceted by farm. Foals transfused with RE HIP at both farms that remained healthy (no pneumonia = 'N') had significantly higher (asterisks represent P<0.05) VapA OD ratios than foals that developed pneumonia (pneumonia = 'Y').

guide for plasma production, ideally leading to improved consistency and quality of plasma produced by manufacturers. Variation in IgG antibody activity in RE HIP both among and within lots of products from each of 3 different commercial manufacturers has been

**Table 1. Odds ratios for the outcome of pneumonia associated with VapA activity-level and farm.**

| Variable | Odds Ratio (95% CI) | P Value |
|---|---|---|
| VapA OD ratio | | |
| High (OD ratio > 0.89) | 1 (NA) | NA |
| Low (OD ratio ≤ 0.89) | 5.95 (2.17–16.13) | 0.000524 |
| Farm | | |
| Farm B | 1 (NA) | NA |
| Farm A | 6.94 (2.49–19.23) | 0.000201 |

Odds ratios (and 95% CIs) estimated using logistic regression for the outcome of pneumonia attributed to *R. equi* among 117 foals transfused with *R. equi* hyperimmune plasma (RE HIP) at both Farm A and Farm B. A binary VapA activity-level variable was created using the median value of VapA of the optical density (OD) ratio for the group of foals was included in the model as well as a variable of farm to account for the potential effects of differences between farms in the method of diagnosis.

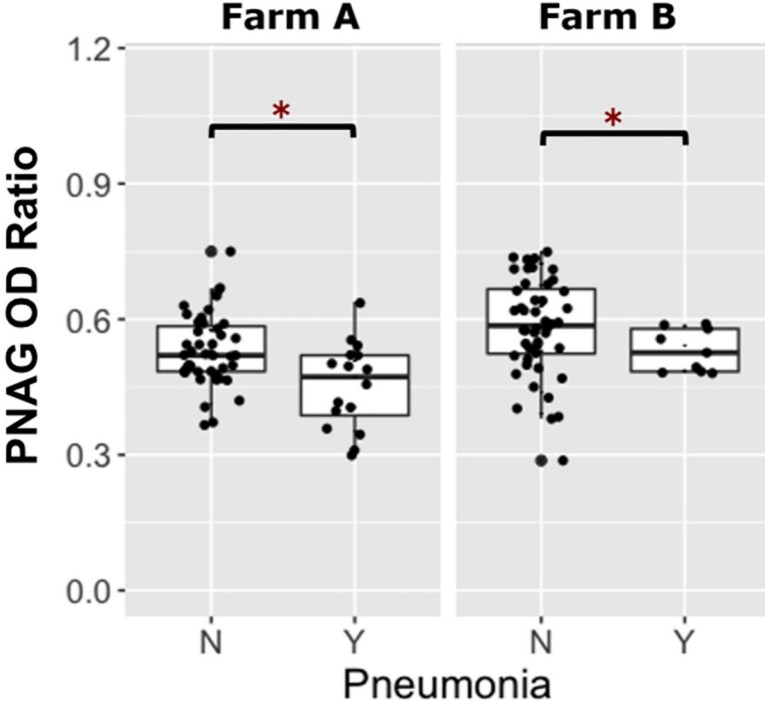

**Fig 5. Foals transfused with PNAG hyperimmune plasma and their comparison of PNAG optical density ratios with whether they developed pneumonia or remained healthy.** Boxplots of optical density (OD) ratios from ELISAs measuring deposition onto purified β-(1→6)-linked poly-*N*-acetyl-glucosamine (PNAG) by serum antibodies in 116 foals transfused with PNAG hyperimmune plasma (PNAG HIP) stratified by pneumonia status, faceted by farm. Foals transfused with PNAG HIP at both farms that remained healthy (no pneumonia = 'N') had significantly higher (asterisks represent P<0.05) PNAG OD ratios than foals that developed pneumonia (pneumonia = 'Y').

documented [34]; the coefficient of variation was as high as 107% for VapA-specific IgGa among lots [34]. In this study, we collected serum samples from transfused foals, but regrettably we did not have samples from the plasma lots post-thawing for transfusion for antibody measurements. Consequently, we cannot differentiate how much of the variability among foals in relative activity of antibodies against VapA, PNAG, and C′1q deposition was attributable to variation among lots of plasma or to other factors such as plasma handling prior to

**Table 2. Odds ratios for the outcome of pneumonia associated with PNAG activity-level and farm.**

| Variable | Odds Ratio (95% CI) | P Value |
|---|---|---|
| PNAG OD ratio | | |
| High (OD ratio > 0.54) | 1 (NA) | NA |
| Low (OD ratio ≤ 0.54) | 2.94 (1.08–8.65) | 0.0347 |
| Farm | | |
| Farm B | 1 (NA) | NA |
| Farm A | 1.49 (0.57–3.91) | 0.4174 |

Odds ratios (and 95% CIs) estimated using logistic regression for the outcome of pneumonia attributed to *R. equi* among 116 foals transfused with β-(1→6)-linked poly-*N*-acetyl-glucosamine hyperimmune plasma (PNAG HIP) at both Farm A and Farm B. A binary PNAG activity-level variable was created using the median value of PNAG optical density (OD) ratio for the group of foals was included in the model as well as a variable of farm to account for the potential effects of differences between farms in the method of diagnosis.

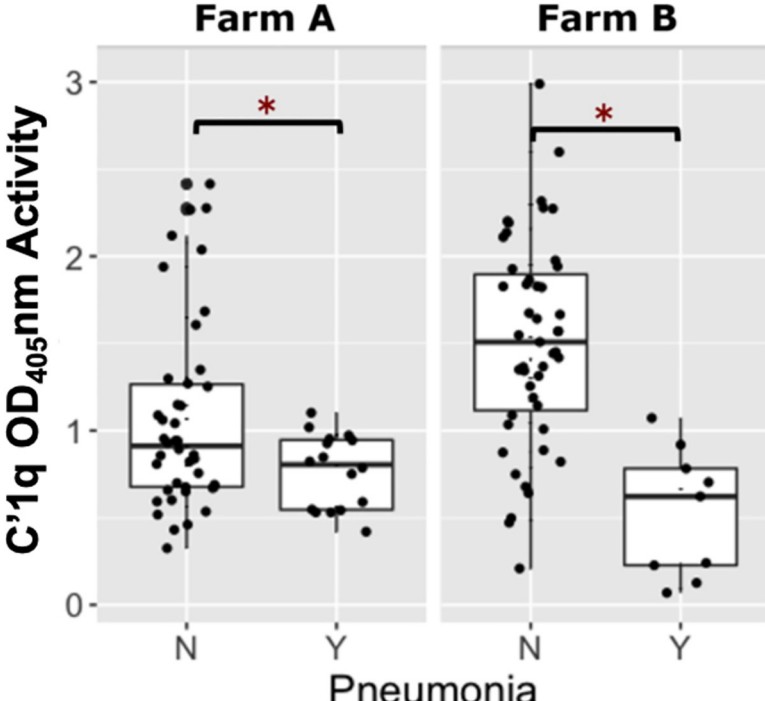

**Fig 6. Foals transfused with PNAG hyperimmune plasma and the comparison of C′1q activity with whether they developed pneumonia or remained healthy.** Boxplot of optical density (OD$_{405}$nm) activities for complement component 1q (C′1q) deposition onto PNAG by serum antibodies in 116 foals transfused with PNAG hyperimmune plasma (PNAG HIP) stratified by pneumonia status, faceted by farm. Foals transfused with PNAG HIP at both farms that remained healthy (no pneumonia = 'N') had significantly higher (asterisks indicate P < 0.05) C′1q deposition activities than foals that developed pneumonia (pneumonia = 'Y').

transfusion, timing of serum sample collection relative to transfusion, and foal-level factors. For example, thawing plasma at too high of a temperature could result in denaturing immunoglobulins. Although we asked farms to collect serum samples immediately post-transfusion, it is possible that there was some variation among foals in the timing of collection that contributed to the observed variation in activity of antibodies among foals. Finally, variability among individual foals in volume of distribution and the background activity of antibodies against

**Table 3. Odds ratios for the outcome of pneumonia associated with C′1q activity-level and farm.**

| Variable | Odds Ratio (95% CI) | P Value |
|---|---|---|
| C′1q OD$_{405}$nm activity Level | | |
| High (OD ratio >1.01) | 1 (NA) | NA |
| Low (OD ratio ≤ 1.01) | 11.37 (3.00–43.48) | 0.0003 |
| Farm | | |
| Farm B | 1 (NA) | NA |
| Farm A | 0.99 (0.35–2.79) | 0.9777 |

Odds ratios (and 95% CIs) estimated using logistic regression for the outcome of pneumonia attributed to *R. equi* among 114 foals transfused with β-(1→6)-linked poly-*N*-acetyl-glucosamine hyperimmune plasma (PNAG HIP) at both Farm A and Farm B. A binary variable for C′1q OD$_{405}$nm relative activity that was created using the median value of C′1q OD$_{405}$nm relative activity for the population of foals as a cut-point was included in the model, as well as a variable of farm to account for the potential effects of differences between farms in the method of diagnosis.

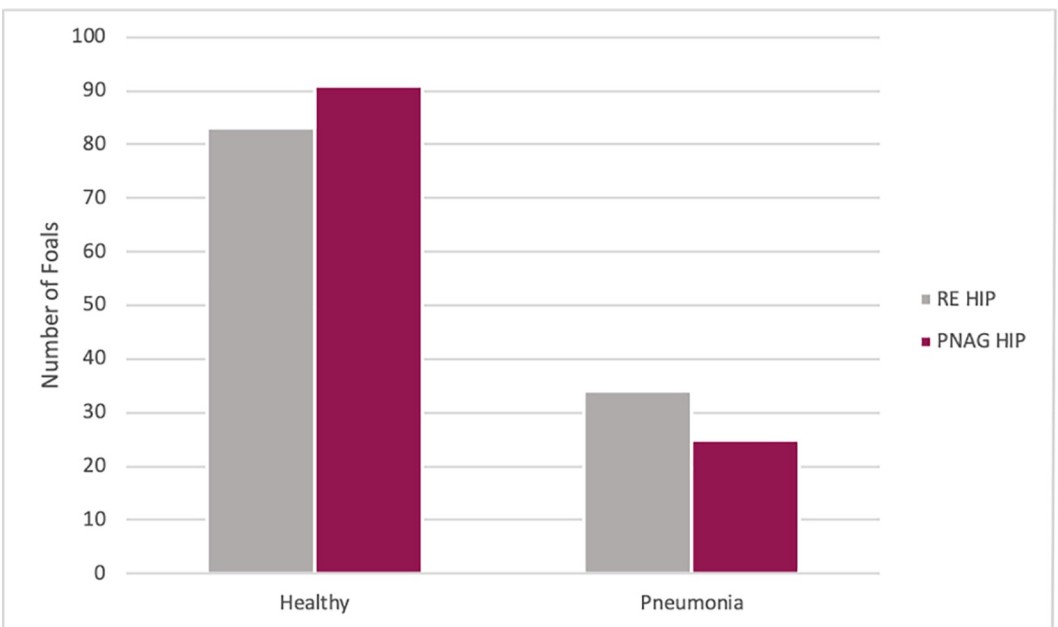

**Fig 7. Distribution of foals that were transfused with *R. equi* hyperimmune plasma or PNAG hyperimmune plasma and whether they developed pneumonia or remained healthy.** The distribution of foals transfused with either *R. equi* hyperimmune plasma (RE HIP), n = 117 foals in grey, and β-1→6-linked poly-*N*-acetyl-glucosamine hyperimmune plasma (PNAG HIP), n = 116 foals in maroon. On the x axis is whether they remained healthy or developed pneumonia. In the RE HIP transfused foals 29% (34/117) foals developed pneumonia compared to the PNAG HIP transfused foals 21% (25/116) foals developed pneumonia.

VapA or PNAG transferred from mares to foals via colostrum could have contributed to the varying OD ratios among foals. This variability in antibody activities among foals, however, enabled us to document that higher values of activity were positively associated with protection against pneumonia in foals.

The finding that antibodies delivered by transfusion can protect foals in an activity-dependent manner suggests that maternal vaccination that results in high colostral levels can be effective for protecting foals against *R. equi* [21,28]. Plasma transfusion will, however, remain an important and commonly practiced method for preventing *R. equi* pneumonia even if a vaccine for *R. equi* pneumonia is developed because not all pregnant mares will be vaccinated

**Table 4. Odds ratios of developing pneumonia associated with type of plasma foal was transfused with and farm.**

| Variable | Odds Ratio (95% CI) | P Value |
|---|---|---|
| Plasma Type | | |
| PNAG HIP | 1 (NA) | NA |
| RE HIP | 1.51 (0.82–2.79) | 0.1832 |
| Farm | | |
| Farm B | 1 (NA) | NA |
| Farm A | 2.82 (1.50–5.32) | 0.0013 |

Odds ratios (and 95% CIs) estimated using logistic regression for the outcome of pneumonia attributed to *R. equi* among 233 foals transfused with either *R. equi* hyperimmune plasma (RE HIP) or β-(1→6)-linked poly-*N*-acetyl-glucosamine hyperimmune plasma (PNAG HIP) at both Farm A and Farm B.

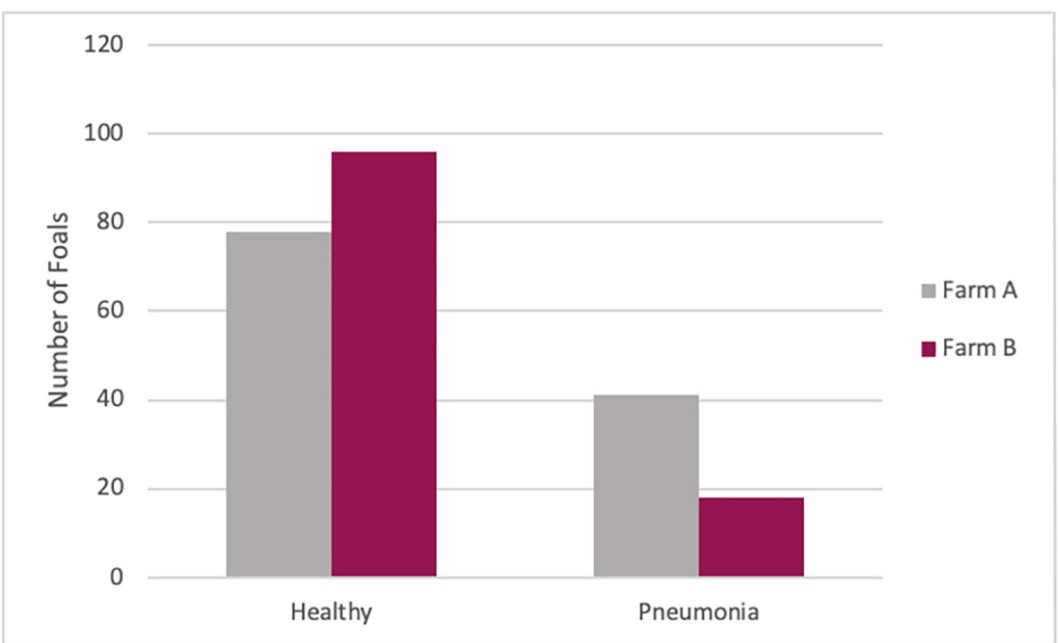

**Fig 8. The distribution of foals that developed pneumonia by Farm.** The distribution of 233 foals that remained healthy or developed pneumonia stratified by Farm on the X axis. At Farm A in grey, 34% (41/119) of foals developed pneumonia and Farm B in maroon, 16% (18/114) of foals developed pneumonia.

or produce high-quality colostrum, and not all foals of vaccinated mares will absorb adequate colostrum.

Among PNAG-transfused foals, activities for C′1q deposition were a stronger predictor of pneumonia than antibody activities against PNAG. C′1 is the initiating protein of the classical complement cascade [38], and it is activated when the immunoglobulins specific to PNAG bind to this portion of the C1qrs molecule [31,38]. Thus, C′1q deposition reflects not merely the amount but the functionality of antibodies. Our findings suggest that the relative levels of functional antibodies measured by assaying C′1q deposition onto PNAG is a better indicator of the potency of plasma than simply measuring anti-PNAG binding activity. It is unclear whether a similar relationship between functional antibody activities versus total antibody activities exists for antibodies against VapA. Interestingly, there was a cluster of foals in the RE HIP group that had relatively high C′1q deposition activities onto the PNAG antigen (Fig 3). Careful review of farm records and comparison of the OD ratios of VapA to C′1q OD activity indicated that these higher-than-expected C′1q OD activities in the RE HIP group were not attributable to labeling or other technical errors in plasma transfusion. None of the RE HIP donors had been vaccinated with PNAG. Because PNAG is found on the surface of many different bacteria, it is possible that this finding is the result of some mares producing functional antibodies against PNAG as a result of infection or natural exposure that were transferred to their foals via colostrum.

The association of a higher activity of antibodies against either VapA or PNAG with reduced odds of pneumonia was observed even after accounting for effects of farm, indicating that transfusion of either plasma protected against both *clinical* and *subclinical* pneumonia. This is consistent with results of a previous observational study [29], and indicates that plasma transfusion has clinical benefits for foals at farms that use ultrasonography or other methods to screen foals for detection of subclinical pneumonia.

The significant effect of farm among foals transfused with RE HIP was attributed to a higher cumulative incidence of pneumonia at Farm A among foals transfused with RE HIP (42%; 25/60) than among foals transfused with PNAG HIP (27%; 16/59), whereas at Farm B the proportion of foals with pneumonia was identical for foals transfused with either RE HIP or PNAG HIP (19%; 9/48). Because pneumonia at Farm A was based on clinical signs whereas pneumonia at Farm B was subclinical, it is possible that PNAG HIP was more effective for protection against clinical than subclinical pneumonia. Further study is needed to substantiate the validity of this observation through replication and to identify any possible mechanism(s) of superior protection against clinical disease for PNAG HIP. Of note, in the study demonstrating protective efficacy of the PNAG vaccine given to mares whose foals were challenged at 4 weeks of life [31], most of the PNAG-immune foals developed ultrasonographic lesions, but only 1 of 12 developed clinical *R. equi* pneumonia. This indicates the antibody to PNAG is highly effective at protecting against disease but not necessarily against subclinical infection based on ultrasonography.

We did not find a significant difference between the 2 plasma products in protection against *R. equi* pneumonia. These results should be interpreted with caution, however, because this study was not designed to test the hypotheses either of superiority or of non-inferiority between these plasma products. Although *in vitro* data indicate PNAG HIP is superior to RE HIP for mediating killing of *R. equi* [32], the study reported here was not designed to compare protection between the 2 plasma types and our absence of evidence of a difference should not be construed as evidence of absence of an effect.

Despite the randomized and masked design, this study had limitations. Not all foals had a trans-endoscopic tracheobronchial aspirate (T-TBA) performed to confirm *R. equi* pneumonia, such that most cases were presumptively diagnosed. However, we do not know of any large breeding farms that perform T-TBA on all foals with suspected *R. equi* pneumonia, and it is unlikely that large breeding farms would consent to this procedure for all foals suspected of *R. equi* pneumonia. Another limitation was that we lacked a placebo or other control group in this study, such as foals that were not transfused or were transfused with plasma from donors that were not hyperimmunized. This was not feasible as the participating farms were unwilling to forego plasma transfusion to foals.

In summary, antibody activities for VapA, PNAG, C′1q are important indicators of protection against *R. equi* pneumonia, and plasma with a higher activity of antibodies against either VapA or PNAG appears more effective for preventing *R. equi* pneumonia.

## Supporting information

**S1 Dataset.**
(CSV)

## Author Contributions

**Conceptualization:** Susanne K. Kahn, Colette Cywes-Bentley, Angela I. Bordin, Daniel R. Vlock, Gerald B. Pier, Noah D. Cohen.

**Data curation:** Susanne K. Kahn, Colette Cywes-Bentley, Glenn P. Blodgett, Nathan M. Canaday, Carly E. Turner-Garcia, Mariana Vinacur, Sophia C. Cortez-Ramirez, Sarah C. Meyer, Gerald B. Pier, Noah D. Cohen.

**Formal analysis:** Susanne K. Kahn, Colette Cywes-Bentley, Mariana Vinacur, Gerald B. Pier, Noah D. Cohen.

**Funding acquisition:** Angela I. Bordin, Noah D. Cohen.

**Investigation:** Susanne K. Kahn, Colette Cywes-Bentley, Glenn P. Blodgett, Nathan M. Canaday, Carly E. Turner-Garcia, Mariana Vinacur, Sophia C. Cortez-Ramirez, Angela I. Bordin, Gerald B. Pier, Noah D. Cohen.

**Methodology:** Colette Cywes-Bentley, Glenn P. Blodgett, Nathan M. Canaday, Carly E. Turner-Garcia, Mariana Vinacur, Angela I. Bordin, Gerald B. Pier, Noah D. Cohen.

**Project administration:** Susanne K. Kahn, Glenn P. Blodgett, Nathan M. Canaday, Carly E. Turner-Garcia, Sarah C. Meyer, Gerald B. Pier, Noah D. Cohen.

**Resources:** Colette Cywes-Bentley, Glenn P. Blodgett, Carly E. Turner-Garcia, Patrick J. Sutter, Sarah C. Meyer, Daniel R. Vlock, Gerald B. Pier, Noah D. Cohen.

**Supervision:** Colette Cywes-Bentley, Glenn P. Blodgett, Nathan M. Canaday, Sarah C. Meyer, Angela I. Bordin, Gerald B. Pier, Noah D. Cohen.

**Validation:** Colette Cywes-Bentley, Nathan M. Canaday, Sarah C. Meyer.

**Writing – original draft:** Susanne K. Kahn, Noah D. Cohen.

**Writing – review & editing:** Susanne K. Kahn, Colette Cywes-Bentley, Glenn P. Blodgett, Nathan M. Canaday, Carly E. Turner-Garcia, Mariana Vinacur, Sophia C. Cortez-Ramirez, Patrick J. Sutter, Sarah C. Meyer, Angela I. Bordin, Daniel R. Vlock, Gerald B. Pier, Noah D. Cohen.

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
