## [Decision Letter · Decision Letter 0]

11 Jun 2021

PONE-D-21-09880

Antibody Activities in Hyperimmune Plasma Against the Rhodococcus equi Virulence -Associated Protein A or Poly-N-Acetyl Glucosamine are Associated with Protection of Foals Against Rhodococcal Pneumonia

PLOS ONE

Dear Dr. Cohen,

Thank you for submitting your manuscript to PLOS ONE. After careful consideration, we feel that it has merit but does not fully meet PLOS ONE’s publication criteria as it currently stands. Therefore, we invite you to submit a revised version of the manuscript that addresses the points raised during the review process.

The expert reviewers appreciate the new findings of the study.  However, a number of clarifications are sought regarding methodology and interpretation.  The response to these is important to further determine the suitability of this manuscript for publication. I urge you to address these issues, should you submit a revised version of manuscript.

We look forward to receiving your revised manuscript.

Kind regards,

Ashlesh K Murthy, M.D., Ph.D.

Academic Editor

PLOS ONE

Journal Requirements:

"I have read the journal's policy and the authors of this manuscript have the following competing interests:

Gerald B. Pier (GBP) is an inventor of intellectual properties [human monoclonal antibody to PNAG and PNAG vaccines] that are licensed by Brigham and Women’s Hospital to Alopexx Vaccine, LLC, an entity in which GBP also holds equity. As an inventor of intellectual properties, GBP also has the right to receive a share of licensing-related income (royalties, fees) through Brigham and Women’s Hospital from Alopexx Vaccine, LLC. GBP’s interests were reviewed and are managed by the Brigham and Women’s Hospital and Partners Health care in accordance with their conflict of interest policies. Colette Cywes-Bentley (CC-B) is an inventor of intellectual properties [use of human monoclonal antibody to PNAG and use of PNAG vaccines] that are licensed by Brigham and Women’s Hospital to Alopexx Vaccine, LLC. As an inventor of intellectual properties, CC-B also has the right to receive a share of licensing-related income (royalties, fees) through Brigham and Women’s Hospital from Alopexx Vaccine, LLC.

Daniel R. Vlock is the Chief Executive Officer of ALOPEXX Vaccines and owns the rights to the PNAG vaccine.  Dr. Vlock had no role in the design or analysis of the study.

SC Meyer and PJ Sutter work for MG Biologics that produced the plasma and thus might have potential earnings; however, they had no part in the design or analysis of the study."

We note that one or more of the authors are employed by a commercial company: 36666 Ranch, Lazy E Ranch, Mg Biologics, Inc, Alopexx Inc.

3.1. Please provide an amended Funding Statement declaring this commercial affiliation, as well as a statement regarding the Role of Funders in your study. If the funding organization did not play a role in the study design, data collection and analysis, decision to publish, or preparation of the manuscript and only provided financial support in the form of authors' salaries and/or research materials, please review your statements relating to the author contributions, and ensure you have specifically and accurately indicated the role(s) that these authors had in your study. You can update author roles in the Author Contributions section of the online submission form.

3.2. Please also provide an updated Competing Interests Statement declaring this commercial affiliation along with any other relevant declarations relating to employment, consultancy, patents, products in development, or marketed products, etc.  

Reviewers' comments:

Reviewer's Responses to Questions

**Comments to the Author**

1. Is the manuscript technically sound, and do the data support the conclusions?

Reviewer #1: Yes

2. Has the statistical analysis been performed appropriately and rigorously? 

Reviewer #1: Yes

3. Have the authors made all data underlying the findings in their manuscript fully available?

Reviewer #1: Yes

4. Is the manuscript presented in an intelligible fashion and written in standard English?

Reviewer #1: Yes

5. Review Comments to the Author

Reviewer #1: The manuscript entitled, “Antibody Activities in Hyperimmune Plasma Against the Rhodococcus equi Virulence -Associated Protein A or Poly-N-Acetyl Glucosamine are Associated with Protection of Foals Against Rhodococcal Pneumonia” aims to assess the impact of two types of hyperimmune plasma on the clinical and subclinical incidence of Rhodococcal pneumonia in foals. My comments are as follows,

1. Do these farms historically use plasma as a preventative measure?

2. Who was responsible for allocating foals to the treatment groups? Was allocation sequence concealed from the person assigning foal to treatment groups? Were any demographics on the foals recorded?

3. Who was responsible for thoracic ultrasound examinations? Who administered the plasma? Were any complications noted following administration of either type of plasma?

4. What was the standard treatment protocol for subclinical/clinical cases?

5. Line -234 how was data that was obtained from this foal before it was lost handled? Was it excluded from the analysis?

6. Line 284 – does the percentage of foals that developed pneumonia reflect those that developed clinical and subclinical pneumonia? Please clarify in the main text and the figure legend.

7. Figure 4: assuming each dot represents an individual animal for Farm B, I count 10 data points for this that developed pneumonia, whereas the text says 9. Please clarify.

8. Figures 7 and 8, please consider labeling the Y-axis.

9. Did all the foals that developed clinical or subclinical survive? Was there any difference in survival for those treated with PNAG HIP vs. RE HIP?

10. Can you please describe the roles of the authors affiliated with Mg Biologics and Alopexx Inc. in the study design, excitation, analysis, and manuscript preparation? Was the plasma provided as gift from these companies?

6. PLOS authors have the option to publish the peer review history of their article (what does this mean?). If published, this will include your full peer review and any attached files.

Reviewer #1: **Yes: **Brina Lopez

---

## [Author Response · Author response to Decision Letter 0]

21 Jun 2021

June 12, 2021

Prof. Ashlesh K. Murthy

Academic Editor

PLoS One

Subject: Revision of manuscript entitled, “Antibody Activities in Hyperimmune Plasma Against the Rhodococcus equi Virulence -Associated Protein A or Poly-N-Acetyl Glucosamine are Associated with Protection of Foals Against Rhodococcal Pneumonia” 

Dear Dr. Murthy:

Thank you for your electronic message dated June 11, 2021 about the above-referenced manuscript. I also thank the reviewers who carefully considered our report. My coauthors and I have revised the report on the basis of the reviewer’s comments that were provided to us. Below, we detail how each point raised by the reviewer was addressed in the revised report. The revised report with marked changes and an unmarked manuscript will be uploaded to the PLoS One website along with this response letter. 

Comments from Reviewer 1:

1. Do these farms historically use plasma as a preventative measure?

AUTHORS’ RESPONSE: We thank the reviewer for pointing out this important omission. The manuscript has been revised to indicate that both farms historically used R. equi hyperimmune plasma transfusion to control R. equi pneumonia. 

2. Who was responsible for allocating foals to the treatment groups? Was allocation sequence concealed from the person assigning foal to treatment groups? Were any demographics on the foals recorded?

AUTHORS’ RESPONSE: We thank the reviewer for raising more great points. Allocation of treatment was made by investigators at Texas A&M University. In the fall of 2018, each farm provided a list of the eligible mares at the farm ordered by expected breeding date. These mares were randomly assigned in a balanced manner to either plasma 0 or 1. The randomly assigned list was provided to the veterinarians at the farm. Because the true foaling date did not always match the expected foaling date, the order of treatment was further randomized by nature. We arrived at this system based on internal discussions with investigators and statisticians and farm personnel. The manuscript has been revised to clarify who was responsible for the treatment allocation and that those administering plasma were aware of which foal was assigned to which plasma. 

We have also revised the report to indicate which additional data (including sex of the Quarter Horse foals) were collected for each foal.

3. Who was responsible for thoracic ultrasound examinations? Who administered the plasma? Were any complications noted following administration of either type of plasma?

AUTHORS’ RESPONSE: The manuscript has been revised to indicate that thoracic ultrasonography was performed by the farm veterinarians (Drs. Blodgett, Canaday, and Turner). The manuscript also has been revised to indicate that plasma transfusion was performed by veterinary technical staff or veterinarians at the farm. The manuscript has been revised to indicate that no adverse effects were noted among transfused foals. We thank the reviewer for prompting us to address each of these important points. 

4. What was the standard treatment protocol for subclinical/clinical cases?

AUTHORS’ RESPONSE: The standard antimicrobial treatment at both farms was a macrolide with or without combination with rifampin. The manuscript has been revised to include these details for each farm. 

5. Line -234 how was data that was obtained from this foal before it was lost handled? Was it excluded from the analysis?

AUTHORS’ RESPONSE: We apologize for the ambiguity. The manuscript has been revised to clarify that the data from this foal were excluded from analysis. 

6. Line 284 – does the percentage of foals that developed pneumonia reflect those that developed clinical and subclinical pneumonia? Please clarify in the main text and the figure legend.

AUTHORS’ RESPONSE: We apologize for the ambiguity. The manuscript has been revised to clarify that the percentage of foals includes those with clinical pneumonia (Farm A) and subclinical pneumonia (Farm B).

7. Figure 4: assuming each dot represents an individual animal for Farm B, I count 10 data points for this that developed pneumonia, whereas the text says 9. Please clarify.

AUTHORS’ RESPONSE: We apologize for the confusion. The extra point was actually an artifact from the same symbol being used to indicate an outlier as was used to reflect individual foals (i.e., an extra dot was added to indicate that foal’s value was an outlier). The figure was revised to correct this problem. 

8. Figures 7 and 8, please consider labeling the Y-axis.

AUTHORS’ RESPONSE: The Y axis has been labelled in revised Figures 7 and 8.

9. Did all the foals that developed clinical or subclinical survive? Was there any difference in survival for those treated with PNAG HIP vs. RE HIP?

AUTHORS’ RESPONSE: We have revised the manuscript to indicate that no study foals died of pneumonia caused by R. equi. The third paragraph from the end of the Discussion indicates that there was no significant difference in protection between the 2 plasma products; however, these results should be interpreted with caution because the study was not designed to test this hypothesis.

10. Can you please describe the roles of the authors affiliated with Mg Biologics and Alopexx Inc. in the study design, excitation, analysis, and manuscript preparation? Was the plasma provided as gift from these companies?

AUTHORS’ RESPONSE: Thank you for these comments. The editor also has asked that this be addressed in the Funding Statement and Competing Interest Statements. Neither Mg Biologics nor Alopexx played a role in the design, execution, analysis, or manuscript preparation for this study EXCEPT that Mg Biologics agreed to make labels masking the plasma identity and to not break the code for the plasma until data collection was completed. Vaccine to hyperimmunize donors against PNAG was provided as a gift by ALOPEXX, Inc. The plasma was purchased by the farms from Mg Biologics. 

Again, we thank you and the reviewer for taking the time to carefully consider our report. Please let us know if you have any additional questions or concerns regarding the revised manuscript.

Sincerely,

Noah D. Cohen

---

## [Editor Report · Decision Letter 1]

16 Aug 2021

Antibody Activities in Hyperimmune Plasma Against the Rhodococcus equi Virulence -Associated Protein A or Poly-N-Acetyl Glucosamine are Associated with Protection of Foals Against Rhodococcal Pneumonia

PONE-D-21-09880R1

Dear Dr. Cohen,

We’re pleased to inform you that your manuscript has been judged scientifically suitable for publication and will be formally accepted for publication once it meets all outstanding technical requirements.

Kind regards,

Ashlesh K Murthy, M.D., Ph.D.

Academic Editor

PLOS ONE
---

## [Editor Report · Acceptance letter]

18 Aug 2021

PONE-D-21-09880R1 

Antibody Activities in Hyperimmune Plasma Against the <i>Rhodococcus equi<i/> Virulence -Associated Protein A or Poly-<i>N<i/>-Acetyl Glucosamine are Associated with Protection of Foals Against Rhodococcal Pneumonia 

Dear Dr. Cohen:

I'm pleased to inform you that your manuscript has been deemed suitable for publication in PLOS ONE. Congratulations! Your manuscript is now with our production department. 

Kind regards, 

on behalf of

Dr Ashlesh K Murthy 

Academic Editor

PLOS ONE